# Total Neoadjuvant Therapy for Locally Advanced Rectal Cancer: Evaluation of Sequencing, Response, and Toxicity in a Single-Institution Cohort

**DOI:** 10.3390/cancers17152416

**Published:** 2025-07-22

**Authors:** Maria Cristina Barba, Paola De Franco, Donatella Russo, Elisa Cavalera, Elisa Ciurlia, Sara De Matteis, Giuseppe Di Paola, Corradino Federico, Angela Leone, Antonella Papaleo, Bianca Santo, Dino Rubini, Giuseppe Rubini, Angela Sardaro

**Affiliations:** 1Radiation Therapy Unit, Department of Onco Hematology, ”Vito Fazzi” Hospital, 73100 Lecce, Italydonatella.russo@asl.lecce.it (D.R.); elisa.ciurlia@asl.lecce.it (E.C.); sara.dematteis@asl.lecce.it (S.D.M.);; 2Department of Palliative Care, ASL Lecce, 73016 San Cesario di Lecce, Italy; corradino.f.federico@gmail.com; 3Radiation Therapy Unit, Department of Precision Medicine, Università degli Studi della Campania Luigi Vanvitelli, 80128 Napoli, Italy; 4Nuclear Medicine Unit, Interdisciplinary Department of Medicine, University of Bari, 70121 Bari, Italy

**Keywords:** total neoadjuvant therapy, locally advanced rectal cancer, organ preservation, pathological complete response, dose intensification radiotherapy

## Abstract

This is a retrospective, single-institution experience with Total Neoadjuvant Therapy in patients diagnosed with Locally Advanced Rectal Cancer. This study aims to contribute to the growing body of evidence by comparing the outcomes of induction chemotherapy, consolidation chemotherapy, and combination (sandwichCHT) within our patient cohort, analyzing the factors associated with pathological complete response, treatment adherence, and toxicity. By analyzing our institutional data, we aim to provide real-world insights into the implementation and outcomes of Total Neoadjuvant Therapy, potentially informing future clinical practice and contributing to the refinement of optimal treatment strategies.

## 1. Introduction

Total neoadjuvant therapy (TNT) has emerged as a promising strategy for managing locally advanced rectal cancer (LARC). This approach aims to optimize oncologic outcomes by administering both chemoradiotherapy (CRT) and systemic chemotherapy (CHT) before surgical resection. Consistent with national and international guidelines [1,2,3], TNT is recommended as the initial treatment for patients with locally advanced low rectal cancer or those at higher risk of local and/or distant metastases. Risk factors identified on magnetic resonance imaging (MRI) include T4 staging, evidence of extramural venous invasion (EMVI), tumor deposits, involvement of the mesorectal fascia (MRF), or a threatened intersphincteric plane.

This comprehensive approach could enhance tumor response, improve treatment compliance, and ultimately increase distant metastasis rate (DM), disease-free survival (DFS), overall survival (OS), and quality of life (QoL).

Pivotal clinical trials, including RAPIDO [4,5], PRODIGE 23 [6,7], STELLAR [8], and OPRA [9,10,11], have demonstrated the potential of TNT to achieve these objectives and facilitate non-operative management (NOM) strategies in selected patients.

The observed pathological complete response (pCR) rate of approximately 20–30% following TNT [12,13] has sparked considerable interest in organ preservation strategies. The observed pathological complete response (pCR) rate of approximately 20–30% following TNT [12,13] interest in organ preservation strategies. These organ-sparing approaches often lead to better bowel function compared to regimens involving neoadjuvant CRT followed by surgery.

Despite its benefits, the optimal sequencing of CHT and CRT within TNT remains a subject of ongoing debate. Specifically, researchers are investigating the comparative effectiveness of induction chemotherapy (iCHT) followed by CRT versus CRT followed by consolidation chemotherapy (cCHT). Trials such as CAO/ARO/AIO-12 [14,15] and OPRA [9,10,11] have provided valuable insights into the trade-offs between treatment compliance and pCR rates associated with these different sequences [16]. Current clinical guidelines [1,2,3] suggest that prioritizing cCHT may be advantageous for patients whose primary objective is local disease control, particularly for those with cT4 disease or MRF involvement. Conversely, iCHT could theoretically be more beneficial for the early management of micrometastases, such as in the presence of EMVI or lymph node-positive disease.

Additionally, the variability in TNT regimens, which includes diverse chemotherapy combinations (triplet vs. doublet) and radiotherapy approaches (long-course CRT [LCRT] vs. short-course radiotherapy [SCRT]), adds complexity to the clinical decision-making process. Beyond these therapeutic considerations, clinical judgment must also carefully assess additional patient-specific attributes, such as age (with elderly patients potentially exhibiting increased susceptibility to TNT-related toxicities), functional status, concurrent medical conditions, and the potential for enduring adverse events following surgery, radiotherapy (RT), and high-dose CHT. Given these multifaceted considerations, there is a pressing need to move beyond a “one-size-fits-all” approach, emphasizing the importance of individualizing treatment plans to optimize patient outcomes and minimize toxicity.

The current challenge lies in accurately identifying patients who achieve an apparent complete response (cCR) to ensure the selection of appropriate candidates for a NOM strategy without compromising oncologic safety. Given the increasing rates of cCR, patient preferences, potential quality of life gains, and the opportunity to avoid surgical morbidity, NOM should be an integral part of the treatment discussion for LARC.

In this context, we present our retrospective, single-institution experience with TNT in patients diagnosed with LARC. This study aims to contribute to the growing body of evidence by comparing the outcomes of iCHT, cCHT, and a combination of both (sandwichCHT) within our patient cohort, analyzing the factors associated with pCR, treatment adherence, and toxicity. By analyzing our institutional data, we aim to provide real-world insights into the implementation and outcomes of TNT, potentially informing future clinical practice and contributing to the refinement of optimal treatment strategies.

## 2. Materials and Methods

This retrospective single-center study analyzed prospectively collected data from our institutional registry of patients with LARC who received TNT between May 2021 and January 2025. The study was conducted following approval from the institutional ethics committee, with informed consent obtained from patients where applicable.

Inclusion criteria were as follows: (a) age ≥ 18 years at the time of diagnosis; (b) histologically confirmed rectal adenocarcinoma; (c) non-metastatic, locally advanced disease, defined as ≥ T3 and/or node-positive status on pre-treatment MRI and computed tomography (CT) scan; (d) post-treatment MRI with diffusion-weighted imaging (DWI) and high-resolution, T2-weighted sequences to assess response; (e) Karnofsky’s Performance Status ≥ 60; (f) white blood cell ≥ 4000 cells/mL; (g) platelet ≥ 100,000 cells/mL.

The exclusion criteria were as follows: metastatic disease, previous CHT, immunotherapy and/or pelvic RT, and other malignancies (coexisting or diagnosed within the last 5 years) except basal cell carcinoma or cervical in situ cancer.

All patients underwent comprehensive baseline staging, including physical examination, rigid or flexible endoscopy, pelvic MRI, and CT of the chest, abdomen, and pelvis. In select cases of diagnostic uncertainty or suspicion of distant disease, integrated Fluorodeoxyglucose Positron Emission Tomography–Computed Tomography (FDG PET-CT) was utilized for improved staging accuracy. Tumor staging was determined according to the 8th edition of the TNM classification. Each case and treatment was reviewed and discussed by our weekly multidisciplinary tumor board (MTB), comprising experienced medical oncologists, radiation oncologists, colorectal surgeons, endoscopists, radiologists, nuclear medicine, and pathologists.

### 2.1. Treatment

Patients received various TNT regimens, including iCHT, cCHT, or a sequential combination of both (sandwichCHT).

For the first time, we chose iCHT for patients at high risk of metastatic disease, specifically those with N2 nodal involvement, positive lateral nodes, or EMVI. Conversely, cCHT was selected for patients with a high risk of local recurrence, identified by cT4 staging, low-located tumors, or MRF involvement. However, a significant challenge arose when patients presented with both metastatic and local recurrence risk factors, complicating the choice of the optimal sequencing strategy. Consequently, we initiated the exploration of a “sandwich chemotherapy” (sandwichCHT) modality.

The decision-making algorithm utilized is summarized in Figure 1.

#### 2.1.1. Chemiotherapy Schedules

The CHT regimens used included the following:-FOLFOX: Oxaliplatin 85 mg/m^2^ intravenously (iv) on day 1, leucovorin 400 mg/m^2^ iv on day 1, 5 fluorouracil-5 FU-400 mg/m^2^ iv as a bolus on day 1, followed by a continuous infusion of fluorouracil 1200 mg/m^2^/day for 2 days. This cycle was repeated every 2 weeks.-CAPEOX: Oxaliplatin 130 mg/m^2^ iv on day 1, and capecitabine 1000 mg/m^2^ orally twice daily for 14 days. This cycle was repeated every 3 weeks.-FOLFIRINOX: Oxaliplatin 85 mg/m^2^ iv on day 1, leucovorin 400 mg/m^2^ iv on day 1, irinotecan 150 mg/m^2^ iv on day 1, and a continuous infusion of 5 FU 1200 mg/m^2^/day for 2 days. This cycle was repeated every 2 weeks.-FOLFOXIRI: Oxaliplatin 85 mg/m^2^ iv on day 1, leucovorin 350 mg/m^2^ iv on day 1, irinotecan 165 mg/m^2^ iv on day 1, and a continuous infusion of 5 FU 1600 mg/m^2^/day for 2 days. This cycle was repeated every 2 weeks.

During the course of LCRT, the concomitant CHT regimen employed consisted of either oral capecitabine (825 mg/m^2^ twice daily on RT days) or intravenous 5 FU (225 mg/m^2^ continuous infusion over 24 h daily for 5 weeks, administered on days 1–5 or days 1–7 of each week of RT). In contrast, for SCRT, CHT was administered solely as part of a consolidation regimen.

#### 2.1.2. Radiotherapy Treatment Planning and Delivery

All patients received RT, either alone or concurrently with CHT.

To individualize treatment strategies based on each patient’s unique anatomy, a comprehensive radiotherapy simulation was performed following our institutional bladder preparation protocol (our protocol advises patients to drink 1.5–2 L of water daily. Before each radiotherapy session, patients are instructed to empty their bladder and then consume 500 mL of water, with treatment commencing 30 min thereafter). This process involved acquiring a non-contrast, enhanced CT scan of the abdomen and pelvis. Given the necessity for consistent patient positioning between the initial simulation and subsequent treatment sessions, careful attention was paid to achieving an optimal setup. All patients were positioned prone on a belly board device to displace the small bowel outside of the intended radiation fields, minimizing potential toxicity. In cases of temporary stoma, the patient’s positioning was carefully selected to ensure accurate treatment reproducibility and maximize patient comfort and compliance.

CT image acquisition was performed with a slice thickness of 3 mm, spanning from the L1 vertebral body to the mid-femur.

Target volume delineation adhered to the guidelines established by Valentini et al. [17].

Two clinical target volumes (CTVs) were defined:-CTV1: This volume encompassed the primary rectal tumor along with the corresponding mesorectum and any pathologically enlarged lymph nodes identified on imaging.-CTV2: This volume consistently included the mesorectum, presacral lymph nodes, obturator lymph nodes, and internal iliac lymph nodes. External iliac and inguinal lymph nodes were incorporated into CTV2 only in the presence of clinically positive findings, extra-mesorectal nodal involvement, or infiltration of the external anal sphincter or the inferior third of the vagina.

A planning target volume (PTV) was generated by expanding the CTV with a margin of 7–8 mm. This expansion accounted for anticipated daily setup variations and organ motion during treatment delivery.

The organs at risk (OARs) considered during treatment planning included the bowel, bladder, femoral heads, and external genitalia.

In the context of LCRT, a total dose of 45 Gy was administered to CTV2, using a fractionation scheme of 1.8 Gy per treatment session. A concomitant or sequential boost was given to CTV1, resulting in a cumulative dose of either 50 Gy or 55 Gy (administered in fractions of 2 Gy or 2.2 Gy, respectively) or 50.4 Gy or 54 Gy in some instances. In contrast, SCRT involved delivering a total dose of 25 Gy at a daily rate of 5 Gy to both CTVs. The choice of dosage and fractionation was made at the discretion of the attending physician to minimize the potential for treatment-related toxicity.

RT was delivered using either an Elekta VERSA-HD linear accelerator employing Volumetric Modulated Arc Therapy (VMAT) or an Elekta PRECISE linear accelerator utilizing static step-and-shot Intensity-Modulated Radiation Therapy (IMRT).

Throughout the entire course of RT, daily patient setup was guided by one sagittal and two lateral skin tattoos in conjunction with lasers. Image-guided radiotherapy (IGRT) using megavoltage cone-beam computed tomography (MV CBCT) was performed twice weekly to verify patient positioning accuracy.

### 2.2. Tolerance and Toxicity

Treatment feasibility and toxicity profiles were closely monitored through weekly clinical assessments and comprehensive laboratory evaluations conducted prior to each iCHT or cCHT cycle, as well as throughout the radiotherapy. Adverse events were documented and graded according to the Common Terminology Criteria for Adverse Events (CTCAE) version 5.0.

Clinical management of adverse events encountered during CHT included the application of supportive therapies and modifications to drug dosing and scheduling, guided by established drug monographs. In cases of RT-related adverse events, supportive measures were employed, with treatment interruption reserved for toxicities reaching Grade ≥ 3.

### 2.3. Treatment Response Evaluation

Instrumental restaging was performed using pelvic MRI with intravenous contrast and CT of the chest, abdomen, and pelvis. These imaging studies were scheduled following the completion of iCHT and again either prior to or following the completion of CRT, as well as before the initiation of cCHT, to serially assess tumor response and identify potential metastatic disease.

A comprehensive pre-surgical clinical and radiological response assessment was conducted 6–8 weeks following the completion of all neoadjuvant therapy. Clinical restaging prior to planned surgical intervention included digital rectal examination (DRE), T2-weighted and DWI MRI of the pelvis [18], and, in carefully selected cases, flexible rectosigmoidoscopy with or without biopsy to evaluate the clinical response within the rectal lumen and determine suitability for surgical resection versus NOM strategies.

The response to TNT was determined by the consensus of the MTB, integrating clinical, endoscopic, and radiological findings, as well as the use of the Response Evaluation Criteria in Solid Tumors (RECIST [19]) version 1.1.

Clinical complete response (cCR) was defined according to established criteria [20,21,22,23] as follows: absence of a palpable mass on DRE, no mucosal irregularity observed on endoscopy, normalization of the rectal wall or minimal residual hypointense thickening on MRI, absence of suspicious regional lymph nodes, and resolution of high signal intensity on b1000 DWI sequences or normalization of the apparent diffusion coefficient (ADC) values at the prior tumor location.

A NOM program was initiated for patients exhibiting a cCR. This comprehensive surveillance strategy involved MRI, endoscopy, and DRE at 3-month intervals for the first 2 years, then at 6-month intervals thereafter. Additionally, CT scans of the chest and abdomen were performed every 6 months for the first 2 years, followed by annual scans thereafter.

Surgical resection, including robotic anterior resection (RAR), abdominoperineal resection (APR) according to Miles’ technique, or local excision (LE), was performed at least 4 weeks following the completion of TNT to allow for maximal treatment effect and resolution of acute toxicities. Total mesorectal excision (TME) was mandatory in both RAR and APR.

Pathological response was assessed on surgical specimens, with pCR defined as the absence of any viable tumor cells (ypT0 ypN0) in both the primary rectal tumor and any resected regional lymph nodes.

Pathological response to TNT, as assessed on surgical specimens, was categorized as follows:-ypCR (pathological complete response);-ypPR (pathological partial response);-ypSD (pathological stable disease).

The pathological response was also classified in accordance with the Tumor Regression Grade (TRG) score [24].

Furthermore, we analyzed key oncological outcomes, including local recurrence rates (LRRs), DFS, and OS, to evaluate the long-term efficacy of the implemented TNT strategies.

## 3. Results

### 3.1. Patients’ Data

Between May 2021 and January 2025, a total of 70 LARC patients were recruited at our institution. The cohort comprised 49 males and 21 females, with a median age of 66 years (range: 35–84).

At initial presentation, the most frequent symptoms were rectal bleeding (77%), altered bowel habits (40%), tenesmus (17%), and abdominal pain (17%). Less common symptoms included mucorrhea (9%) and symptoms suggestive of subocclusion or occlusion (1%).

All patients underwent colonscopy, which confirmed a histopathological diagnosis of adenocarcinoma. The tumors were located in the upper rectum (more than 10.1 cm from the internal anal orifice—IAO) in 12 patients (17%), in the middle rectum (between 6.1 and 10 cm from IAO) in 26 patients (37%), and in the lower rectum (less than 6 cm from IAO) in 32 patients (46%).

Preoperative staging, based on the 8th edition of the TNM classification, was primarily clinical, utilizing pelvic MRI and CT of the chest, abdomen, and pelvis. One patient declined MRI due to claustrophobia. Further detailed local staging was achieved with echoendoscopy in 24 patients (34%), while 7 patients (10%) underwent FDG PET-CT for systemic staging and/or assessment of equivocal findings.

The majority of the cohort was classified as clinical T3 (73%) and/or clinical N2 (70%). MRF involvement in imaging was identified in 49% of cases.

The detailed clinical characteristics of the enrolled LARC patient cohort are summarized in Table 1.

### 3.2. Chemotherapy Regimens

A total of 70 patients were included in this analysis and received neoadjuvant chemotherapy. Of these, 29 patients underwent iCHT, 7 received cCHT, and 34 were treated with sandwichCHT. Prior to the initiation of CHT, all patients underwent placement of a peripherally inserted central catheter (PICC), and dihydropyrimidine dehydrogenase (DPYD) enzyme activity was assessed in order to individualize chemotherapy dosage, in case of DPYD genetic mutation. The specific CHT regimens administered varied (Table 2).

#### 3.2.1. Induction Chemotherapy (iCHT):

-FOLFOX: 6 patients received 3 to 8 cycles of FOLFOX;-CAPEOX: 21 patients received 2 to 6 cycles of CAPEOX;-FOLFIRINOX: 2 patients received 5 to 6 cycles of FOLFIRINOX.

#### 3.2.2. Consolidation Chemoradiotherapy (cCHT):

-FOLFOX: 1 patient received 4 cycles of FOLFOX;-CAPEOX: 2 patients received 3 to 4 cycles of CAPEOX;-FOLFOXIRI: 4 patients received 8 cycles of FOLFOXIRI.

#### 3.2.3. Sandwich Chemotherapy (sandwichCHT):

-FOLFOX: 3 patients received 4 to 6 cycles of induction FOLFOX followed by 2 to 4 cycles of consolidation FOLFOX, for a total of 8 cycles of FOLFOX;-FOLFOXIRI: 1 patient received 5 cycles of induction FOLFOXIRI followed by 3 cycles of consolidation FOLFOXIRI, for a total of 8 cycles of FOLFOXIRI;-CAPEOX: 30 patients received 2 to 5 cycles of induction CAPEOX followed by 2 to 4 cycles of consolidation CAPEOX, for a total of 6 cycles of CAPEOX.

#### 3.2.4. Concomitant Chemotherapy

Of the 66 patients (94%) undergoing LCRT, 64 received concurrent CHT, consisting of either oral capecitabine (58 patients, 91%), or intravenous 5 FU (6 patients, 9%).

### 3.3. Radiotherapy

Various fractionation schedules were employed:-50 patients received 45 Gy to the CTV2 and 55 Gy to the CTV1 using a SIB with a hypofractionation regimen (1.8 and 2.2 Gy per fraction, respectively). In 14 patients (28%), pathological extra-mesorectal nodals were included in CTV1;-6 patients received 45 Gy to the CTV2 and 50 Gy to the CTV1 using a simultaneous integrated boost (SIB) with a conventional fractionation regimen (1.8 and 2 Gy per fraction, respectively);-5 patients received 45 Gy to the CTV2 and a sequential boost to CTV1 of 9 Gy (total dose: 54 Gy), delivered with a conventional fractionation regimen of 1.8 Gy per fraction;-5 patients received 45 Gy to the CTV2 and a sequential boost to CTV1 of 5.4 Gy (total dose: 50.4 Gy), delivered with a conventional fractionation regimen of 1.8 Gy per fraction;-4 patients received 25 Gy to both CTV1 and CTV2 using an ultra-hypofractionated regimen of 5 Gy per fraction (total dose: 25 Gy).

Detailed characteristics of the specific treatment regimens are summarized in Table 3.

### 3.4. Treatment-Related Toxicities

During iCHT, one patient (3%) experienced Grade 4 gastrointestinal toxicity, and one patient (3%) experienced Grade 3 hematological toxicity. Grade 2 hematological toxicity occurred in three patients (10%), Grade 2 gastrointestinal toxicity in two patients (7%), Grade 2 skin toxicity in one patient (3%), and peripheral neuropathy in one patient (3%). No genitourinary toxicity was observed. One patient experienced an acute myocardial infarction one month following the completion of iCHT. CHT was discontinued in one case (3%), reduced to 75% of the original dose in four cases (14%), and reduced to 50% in one case (3%).

During cCHT, no Grade 4 or Grade 3 toxicities, nor any skin, genitourinary, or cardiac toxicities were reported. One patient (14%) experienced Grade 2 hematological toxicity, one patient (14%) experienced Grade 2 gastrointestinal toxicity, and three patients (43%) developed peripheral neuropathy. No changes in the dosage or timing of administration were necessary.

During the induction phase of sandwichCHT, one patient (3%) experienced Grade 3 gastrointestinal toxicity, one patient (3%) experienced Grade 2 hematological toxicity, three patients (9%) experienced Grade 2 gastrointestinal toxicity, and four patients (12%) developed peripheral neuropathy. No genitourinary or cardiac toxicities were observed. In two cases (6%), one of the two chemotherapy drugs was discontinued, and in two cases (6%), the dosage was reduced to 75% of the original amount. The timing of administration was changed in one case (3%).

During the consolidation phase of sandwichCHT, no Grade 4 or Grade 3 toxicities were reported. One patient (3%) experienced Grade 2 gastrointestinal toxicity, one patient (3%) experienced Grade 2 genitourinary toxicity, and two patients (6%) developed peripheral neuropathy. No changes in the dosage or timing of administration were necessary.

During the course of CRT, 1 patient (2%) experienced Grade 3 hematological toxicity, 1 patient (2%) experienced Grade 2 gastrointestinal toxicity, 3 patients (6%) experienced Grade 2 hematological toxicity, 23 patients (35%) experienced Grade 2 gastrointestinal toxicity, 8 patients (12%) experienced Grade 2 genitourinary toxicity, and 2 patients (3%) experienced Grade 2 skin toxicity. The dosage of concomitant CHT was reduced to 75% in five cases (8%) and to 50% in one case (2%). Radiotherapy was interrupted for more than five fractions in only four patients.

Table 4 provides a comprehensive overview of the type and grade of toxicities experienced by patients within each treatment regimen.

### 3.5. Response to TNT

Following the completion of TNT, all patients underwent comprehensive restaging, which included pelvic MRI and CT scans of the chest, abdomen, and pelvis. The clinical TNM stage after TNT was assessed by MTB according to RECIST v1.1. The observed responses were as follows:-Complete response (ycCR) was documented in 23 patients (33%). Of these, 15 patients (65%) underwent sandwichCHT, 5 patients (22%) underwent iCHT, and 3 patients (13%) underwent cCHT.-Major response (ycMR) was observed in 28 patients (40%). Among these, 13 patients (46%) received sandwichCHT, 13 patients (46%) received iCHT, and 2 patients (8%) received cCHT.-Stable disease (ycSD) was reported in 14 patients (20%). Among them, 5 patients (38%) underwent sandwichCHT, 6 (46%) patients underwent iCHT, and 2 patients (16%) underwent cCHT.-Progression disease (ycPD) was identified in 5 patients (7%). These patients received sandwichCHT in 2 cases (40%) and iCHT in 3 cases (60%).

### 3.6. Surgery or NOM

Surgical intervention was performed in 60 patients (86%), with a median interval of 8 weeks (median 60.5 days, range 15–282 days) post-completion of TNT. A RAR was executed in 55 patients (92%), with a temporary diverting loop ileostomy in 34 patients (63%). Subsequent ileostomy reversal was performed after a median of 48 days (range 2–315). An APR, according to Miles’ technique, was performed in 4 patients (7%). Local excision was performed in one patient (1%). Pathological evaluation of all resected specimens (100%) confirmed complete resection with microscopically negative margins (R0 resection).

Postoperative complications included anastomotic leakage in 6 patients (10%), pelvic abscess formation in 3 patients (5%), and small bowel obstruction (subocclusion) in 1 patient (1%).

Of the 5 patients (7%) exhibiting disease progression during TNT, 3 initiated first-line systemic CHT, while 2 underwent surgical resection of the primary tumor.

Five patients (7%) demonstrating ycCR at restaging were enrolled in a NOM protocol involving close surveillance (“watch-and-wait”).

Pathological response to TNT resulted in the following:-ypCR: Observed in 18 patients (30%). Subgroup analysis revealed that 10 patients (6%) received sandwichCHT, 6 patients (3%) received iCHT, and 2 patients (1%) received cCHT.-ypPR: Observed in 20 patients (33%). Among patients with ypPR, 8 patients (40%) were classified as pT1 (3 received sandwichCHT, 4 received iCHT, 1 received cCHT), and 12 patients (60%) were classified as pT2 (7 received sandwichCHT, 5 received iCHT).-ypSD: Observed in 22 patients (37%). Among patients with ypSD, 7 patients (32%) received sandwichCHT, 11 patients (50%) received iCHT, and 4 patients (18%) received cCHT.

TRG analysis revealed the following distribution:-TRG 1 (minimal residual disease): Observed in 19 patients (32%), of whom 10 (53%) received sandwichCHT, 6 (32%) received iCHT, and 3 (15%) received cCHT.-TRG 2 (moderate residual disease): Observed in 25 patients (42%), of whom 10 (40%) received sandwichCHT, 13 (52%) received iCHT, and 2 (8%) received cCHT.-TRG 3 (extensive residual disease): Observed in 12 patients (20%), of whom 5 (42%) received sandwichCHT, 5 (42%) received iCHT, and 2 (16%) received cCHT.-TRG 4 (no regression/predominant residual disease): Observed in 4 patients (6%), of whom 2 (50%) received sandwichCHT and 2 (50%) received iCHT.

Table 5 provides a comparison between the clinical response evaluated by MTB and the corresponding pathological response obtained from surgical specimens.

### 3.7. Follow-Up and Oncological Outcomes

Following surgical resection, all patients underwent clinical and instrumental follow-up (FUP) in accordance with national and international guidelines. The NOM patient cohort followed a more intensive FUP, with evaluation every 3 months for the initial 2 years, according to institutional protocol.

To date, after a median FUP duration of 17 months (range: 6 weeks–41 months), 65 patients (93%) remained alive and without evidence of local recurrence (LRR 0%). Post-surgery, 10 patients (17%) experienced disease progression (PD). Specifically, 1 patient developed peritoneal metastasis 15 months post-surgery, 4 patients were diagnosed with hepatic metastasis after a median FUP of 12 months, 4 patients developed lung metastasis after a median FUP of 16.5, and 1 patient was diagnosed with bone metastasis at 3 months post-surgery. The median DFS was 14.7 months.

In a subgroup analysis of these patients, 4 received sandwichCHT, 5 iCHT, and 1 cCHT.

At the time of analysis, 55 patients (79%) were disease-free, 5 patients (7%) had controlled disease, and 5 patients (7%) had uncontrolled disease. Five patients (7%) had died. Of these deaths, three patients had not undergone surgery due to progression during TNT, one patient experienced systemic progression after surgery, and one patient died due to COVID-19 infection.

The median OS, calculated as the time from the date of diagnosis to the date of last FUP or death, was 26 months.

## 4. Discussion

TNT represents a strategic intensification of preoperative management for LARC, employing a sequential approach of both RT and CHT prior to surgical resection. The increasing adoption of TNT signifies a potential paradigm shift in LARC management, driven by its promise of enhanced short-term efficacy, evidenced by higher rates of pCR and R0 resection. Furthermore, TNT is hypothesized to translate into improved long-term oncological outcomes, including OS, DFS, and distant metastasis-free survival (DMFS), in comparison to the conventional neoadjuvant CRT followed by radical surgery sequence [4,6,25,26].

The heterogeneity of the proposed TNT protocols for high- to very-high-risk LARC is notable, with variations in patient selection criteria based on local disease extent and diverse sequencing of systemic therapy and (C) RT. Current studies have not demonstrated statistically differences in DFS, OS, sphincter-saving surgery rates, R0 resection rates, or postoperative complications across these various TNT regimens [5,8,27,28,29,30,31,32,33].

Two primary approaches have emerged: iCHT followed by LCRT, and either LCRT or SCRT, followed by cCHT. Each of these strategies presents distinct theoretical advantages and potential drawbacks. The decision regarding induction versus consolidation chemotherapy within a TNT framework necessitates careful individualization, considering the specific clinical and tumor characteristics of each patient [34].

Induction TNT provides the theoretical advantage of early intervention against micrometastatic disease. However, a potential concern is the risk of selecting chemoresistant clones, potentially diminishing the efficacy of subsequent CRT. Several studies have indicated a more than twofold-increased odds of major adverse events with induction TNT compared to standard neoadjuvant CRT, particularly concerning hematologic, gastrointestinal, and neuropathic toxicities. While often manageable with supportive care and dose modifications, these toxicities can significantly impact treatment adherence and patient quality of life [35].

Conversely, data from pivotal trials such as CAO/ARO/AIO-12 [15] and OPRA [11], along with studies by Ma [29] and Wu [36], suggest that patients receiving CRT followed by cCHT achieve higher pCR rates compared to those treated with iCHT. This observation may be attributed to the longer interval between the completion of LCRT and surgical intervention, which could enhance the RT effect. These findings hint at a potential advantage of cCHT over iCHT in terms of organ preservation strategies.

Consistent with these observations and the recent American Society of Clinical Oncology (ASCO) guidelines, a conditional recommendation with moderate evidence quality suggests that for TNT candidates, chemotherapy should ideally follow radiation therapy [2].

In our single-center experience, the sandwichCHT regimen, a sequential combination of induction and consolidation chemotherapy flanking LCRT, was the predominant approach (49% utilization). We reported a pathological complete response (pCR) rate of 30%, consistent with the published data. Notably, 56% of patients achieving CR were treated with the sandwichCHT regimen. By incorporating patients with clinical complete response managed with a NOM protocol, our overall complete response rate increased to 33%.

The potential mechanisms underlying TNT’s observed benefits, such as earlier intervention against micrometastatic disease, decreased tumor cell activity and circulating tumor cells, and mitigation of metastasis and implantation risks associated with local therapies, may contribute to improved OS, DMFS, and consequently, DFS. In our study, with a median follow-up of 17 months (range: 6 weeks–41 months), we reported a median OS of 26 months and a DFS of 14.7 months. Notably, we achieved complete local control, with a LRR of 0%. We attribute this favorable local control primarily to the advanced radiotherapy techniques employed, as all patients received either VMAT or IMRT, with 78% receiving a boost dose exceeding 50.4 Gy.

Despite the prolonged chemotherapy duration inherent in TNT regimens and the cumulative adverse effects from both RT and CHT, the incidence of Grade 3–4 toxicity in TNT is generally reported as higher than standard CRT. However, this increased risk does not appear to negatively impact the observed excellent compliance rates and long-term survival outcomes in TNT cohorts [29,37]. In our cohort, TNT was well tolerated; we noted one case of Grade 4 gastrointestinal toxicity (in the iCHT arm) and two instances of Grade 3 toxicities (one hematological in the iCHT arm and one during CRT). Grade 2 gastrointestinal toxicity during CRT was the most common (33%). The predominant use of sandwichCHT across our cohort suggests that this sequence may be better tolerated, potentially due to the split chemotherapy and the moderate-dose intensified radiotherapy (median 55 Gy) applied in the majority (71%) of cases.

The trials analyzed [4,6,8,14,38] employed either conventional fractionation schedules for RT (50 Gy in 25 fractions or 50.4 Gy in 28 fractions) or ultra-hypofractionated regimens (25 Gy in 5 fractions). Recent data indicate that moderate-dose intensification (52.5–57.5 Gy) with concurrent full-dose capecitabine is associated with low rates of Grade 3 acute toxicity, high patient compliance, and encouraging downstaging outcomes [39,40,41,42,43].

Regarding surgical outcomes, R0 resection rates have generally been higher in TNT groups compared to standard CRT. Interestingly, only the Polish trial [38] reported a non-significant trend towards a lower R0 resection rate in the TNT arm (71% vs. 77%, *p* = 0.07). In our series, we achieved complete (R0) resection in all cases (100%).

Current evidence suggests that TNT does not increase postoperative morbidity rates compared to standard CRT [5,7,8]. In our study, postoperative complications were observed in 10 patients (16%), including anastomotic leaks, pelvic abscess, and small bowel obstruction.

Our study is subject to several limitations. The analysis of three distinct TNT schedules with varying CHT types and cycles complicates data interpretation and comparison. These variations in systemic therapy and RT protocols may serve as confounding variables affecting the outcomes observed. Furthermore, the limited median follow-up duration of 17 months precludes definitive conclusions regarding long-term survival advantages. Finally, the inclusion of survival data from both radically resected and non-operatively managed patients introduces further heterogeneity. The lack of standardized research protocols in this field underscores the ongoing challenge of definitively identifying the optimal TNT strategy and the patient populations that derive the greatest benefit.

Concomitantly, the identification of predictive biomarkers—encompassing clinical, genetic, radiomic, and immune parameters—is crucial for personalizing treatment approaches for individual patients [44,45,46,47].

## 5. Conclusions

In our single-center experience employing a TNT protocol that incorporates intensified radiotherapy, we observed encouraging short-term efficacy, evidenced by a pCR rate of 30%. This was accompanied by a generally favorable tolerability profile predominantly characterized by moderate adverse events. Notably, the observed association between the sandwichCHT regimen and both a higher pCR rate and a manageable toxicity profile warrants further investigation in prospective, multi-center studies with extended follow-up.

Such research is essential to confirm whether this balance of efficacy and tolerability translates into sustained long-term oncological outcomes, including OS and disease control in LARC, and to refine patient selection criteria for this intensive neoadjuvant approach.

## Figures and Tables

**Figure 1 cancers-17-02416-f001:**
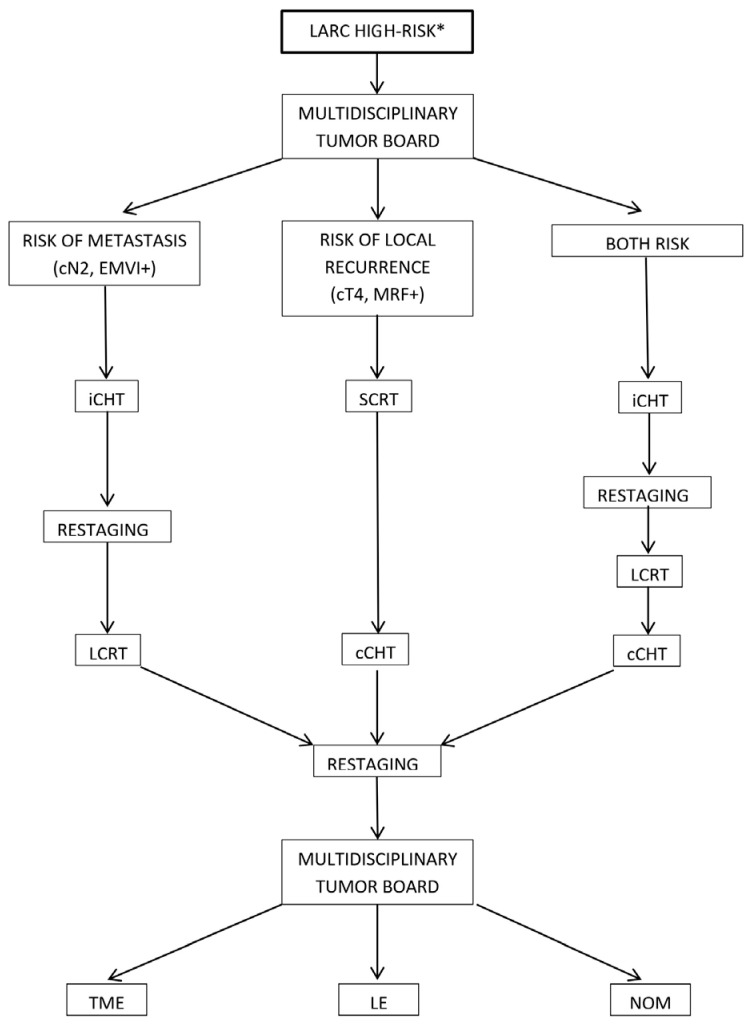
Decision-making algorithm for total neoadjuvant therapy (TNT) sequencing and patient management. * cT4, cN2, EMVI, tumor deposits, MRF involvement, and a threatened intersphincteric plane. cCHT: consolidation chemotherapy, iCHT: induction chemotherapy, LARC: locally advanced rectal cancer, LCRT: long-course CRT, LE: local excision, MRF: mesorectal fascia, NOM: non-operative management, SCRT: short-course radiotherapy, TME: total mesorectal excision.

**Table 1 cancers-17-02416-t001:** Clinical characteristics of the patients enrolled in the study.

Characteristics	N°
Median Age (range)	66 years (range: 35–84)
Sex	
- Male	49 (70%)
- Female	21 (30%)
Comorbidity (alone or mixed)	
- None	33 (47%)
- Hypertension	24 (34%)
- Hyperlipidemia	10 (14%)
- Diabetes	9 (13%)
- Colon’s diverticula	8 (11%)
- Heart disease	5 (7%)
- Chronic Obstructive Pulmonary Disease (COPD)	2 (3%)
Symptoms (alone or mixed)	
- Bleeding	54 (77%)
- Irregular alvus	40 (57%)
- Tenesmus	12 (17%)
- Abdominal pain	12 (17%)
- Mucorrhoea	6 (9%)
- Subocclusion/Occlusion	1 (1%)
Tumor location	
- Upper	12 (17%)
- Middle	26 (37%)
- Lower	32 (46%)
Quadrants involved	
- 1	4 (6%)
- 2	16 (23%)
- 3	22 (31%)
- 4	28 (40%)
Cranial-caudal extension (mm)	
- <30	4 (6%)
- 31–60	35 (50%)
- 61–90	26 (37%)
- >91	5 (7%)
MRF positive	
- No	36 (51%)
- Yes	34 (49%)
Clinical T stage	
- cT2	1 (1%)
- cT3	51 (73%)
- cT4	18 (26%)
Clinical N stage	
- cN0	3 (4%)
- cN1	18 (26%)
- cN2	49 (70%)

**Table 2 cancers-17-02416-t002:** Chemotherapy regimens.

Chemotherapy Regimens	N ° (%)
*Induction*	29 (41%)
- FOLFOX	6 (21%)
- CAPEOX	21 (72%)
- FOLFIRINOX	2 (7%)
*Consolidation*	7 (10%)
- FOLFOX	1 (14%)
- CAPEOX	2 (28%)
- FOLFOXIRI	4 (58%)
*Sandwich*	34 (49%)
- FOLFOX	3 (9%)
- CAPEOX	30 (88%)
- FOLFOXIRI	1 (3%)
*Concomitant*	66 (94%)
- Capoecitabine	60 (91%)
- 5 FU	6 (9%)

**Table 3 cancers-17-02416-t003:** Radiotherapy treatment.

Radiotherapy Schedules	N° (%)
Long course: 45 Gy to CTV2 and 55 Gy to CTV1, simultaneous integrated boost	50 (71%)
Long course: 45 Gy to CTV2 and 50 Gy to CTV1, simultaneous integrated boost	6 (9%)
Long course: 45 Gy to CTV2 and 54 Gy to CTV1, sequential boost	5 (7%)
Long course: 45 Gy to CTV2 and 50.4 Gy to CTV1, sequential boost	5 (7%)
Short course: 25 Gy to both CTV1 and CTV2	4 (6%)

**Table 4 cancers-17-02416-t004:** Summary of treatment toxicities.

Toxicity Grade	Regimen	Type of Toxicity	Number of Patients Affected	Total for Grade
Grade 4	iCHT	Gastrointestinal	1	1
Grade 3	iCHT	Hematological	1	2
	CRT	Hematological	1	
Grade 2	iCHT	Hematological	3	36
	iCHT	Gastrointestinal	2	
	iCHT	Skin	1	
	cCHT	Hematological	1	
	cCHT	Gastrointestinal	1	
	sandwichCHT (Induction)	Hematological	1	
	sandwichCHT (Induction)	Gastrointestinal	3	
	sandwichCHT (Consolidation)	Gastrointestinal	1	
	sandwichCHT (Consolidation)	Genitourinary	1	
	CRT	Hematological	3	
	CRT	Gastrointestinal	23	
	CRT	Genitourinary	8	
	CRT	Skin	2	

**Table 5 cancers-17-02416-t005:** Comparison between the clinical response evaluated by MTB and the corresponding pathological response obtained from surgical specimens.

Clinical Response (70 Patients)	Pathological Response (60 Patients)	
ycCR: 23 (33%)	ypCR: 20 (33%)	TRG 1: 19 (32%)
ycMR: 28 (40%)	ypPR: 20 (33%)	TRG 2: 25 (42%)
ycSD: 14 (20%)	ypSD: 22 (37%)	TRG 3: 12 (20%)
ycPD: 5 (7%)		TRG 4: 4 (6%)

## Data Availability

The data presented in this study are available on request from the corresponding author due to privacy.

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
