# Peer review of "Total Neoadjuvant Therapy for Locally Advanced Rectal Cancer: Evaluation of Sequencing, Response, and Toxicity in a Single-Institution Cohort"

_cancers, 2025, doi:10.3390/cancers17152416_

Round 1

Reviewer 1 Report

Comments and Suggestions for Authors

Authors have presented a thorough investigation in the study “Total Neoadjuvant Therapy for Locally Advanced Rectal Cancer: Evaluation of Sequencing, Response, and Toxicity in a Single-Institution Cohort”. The current study is well designed written and technically meaningful. However, there are some areas that require clarifications before acceptance of the article for publication. While the current study shows the real-world data but has significant limitations related to its design, sample size, heterogeneity, and follow-up time. I advise the editor acceptance with minor revision of the following comments

Scientific/Major

  1. How did you consider the optimal sequencing of chemotherapy and CRT (induction versus consolidation versus sandwich) in TNT for LARC?
  2. A handful of chemotherapy regimens (e.g. FOLFOX, CAPEOX, FOLFIRINOX) was used in treatment. Did author observe any impact of dosage regimen heterogeneity in between chemo and RT which might lead to difference in efficacy or toxicity?
  3. Does the author observe any positive or negative impact compared with standard CRT treatment with surgery outside of TNT approach?
  4. What was the election criteria for selecting patients for TNT approach? What is the impact of patient background on outcomes of therapy and toxicity? Also urinary/bowel function data on the patients are missing. Can selection of patients based on these criteria impact the study outcomes?
  5. The median follow-up time was 17 months. Gastrointestinal cancers often need longer follow up times due to late recurrence/remission, survival. What is the median follow up time used in literature and does the short follow-up time would change the conclusion incase of a longer follow up time?

Minor

  1. Please correct the spelling mistake on line 60 ‘wich’.
  2. Please remove one ‘a’ from line 96.
  3. Please correct the spelling of ‘recived’ to ‘received’ throughout the text.

Author Response

Comments 1: Authors have presented a thorough investigation in the study “Total Neoadjuvant Therapy for Locally Advanced Rectal Cancer: Evaluation of Sequencing, Response, and Toxicity in a Single-Institution Cohort”. The current study is well designed written and technically meaningful. However, there are some areas that require clarifications before acceptance of the article for publication. While the current study shows the real-world data but has significant limitations related to its design, sample size, heterogeneity, and follow-up time. I advise the editor acceptance with minor revision of the following comments.

How did you consider the optimal sequencing of chemotherapy and CRT (induction versus consolidation versus sandwich) in TNT for LARC?

Response 1: Thanks for your question. Initially, we individualized TNT sequencing based on risk profiles for LARC. We chose induction chemotherapy for patients at high risk of metastatic disease, and consolidation chemotherapy was selected for patients with a high risk of local recurrence. However, a significant challenge arose when patients presented with both metastatic and local recurrence risk factors, complicating the choice of an optimal sequencing strategy. Consequently, we began exploring a "sandwich chemotherapy" modality. Given its low toxicity and good tolerability, this approach has become our preferred treatment. While it's not currently included in any European guidelines and, to our knowledge, no other institutions have yet investigated this specific sequencing, we hope our paper will stimulate further research into this promising regimen.

We've updated the manuscript to include our decision-making algorithm and Figure 1. These revisions can be found in the Materials and Method section.

Comments 2: A handful of chemotherapy regimens (e.g. FOLFOX, CAPEOX, FOLFIRINOX) was used in treatment. Did author observe any impact of dosage regimen heterogeneity in between chemo and RT which might lead to difference in efficacy or toxicity?

Response 2: Thanks for your comment. We observed good tolerance and a low toxicity profile across all chemotherapy schedules used. Consistent with same chemoradiotherapy, we noted a higher frequency of toxicity with induction chemotherapy when CAPEOX was administered. However, we believe our current patient cohort size is too small to draw robust conclusions regarding the impact of dosage regimen heterogeneity on efficacy or toxicity.

Comments 3: Does the author observe any positive or negative impact compared with standard CRT treatment with surgery outside of TNT approach?

Response 3: That's an interesting question. We observed fewer gastrointestinal toxicities with the TNT approach. Splitting chemotherapy treatment using a "sandwich" approach appeared to be better tolerated than a full induction regimen. Our study also showed an improved pCR rate with TNT compared to neoadjuvant CRT alone, which aligns with published literature.

Comments 4: What was the election criteria for selecting patients for TNT approach? What is the impact of patient background on outcomes of therapy and toxicity? Also urinary/bowel function data on the patients are missing. Can selection of patients based on these criteria impact the study outcomes?

Response 4: Thank you for highlighting these important points. We've enhanced the Introduction to address your valuable feedback. Specifically, we now include:

•        A more explicit discussion of national and international guidelines recommending TNT, along with insights into the ongoing debate regarding optimal sequencing within TNT regimens. These revisions can be found in the Introduction section.

•        A clearer articulation of the need for individualized treatment strategies, reflecting the complex interplay of patient and treatment factors. These revisions can be found in the Introduction section.

•        Our decision-making algorithm (Figure 1) and details regarding our clinical management of adverse events. These revisions can be found in the Materials and Method section.

Comments 5: The median follow-up time was 17 months. Gastrointestinal cancers often need longer follow up times due to late recurrence/remission, survival. What is the median follow up time used in literature and does the short follow-up time would change the conclusion in case of a longer follow up time?

Response 5: We agree that a longer follow-up period is often crucial for gastrointestinal cancers due to the potential for late recurrence and to fully assess long-term survival and remission. Major TNT trials, such as RAPIDO and PRODIGE-23, have reported median follow-up times in the range of 46.5 to 64 months. The present follow-up duration of 17 months in our study, while sufficient for initial insights, indeed precludes more comprehensive or definitive long-term analyses and results. We plan to provide an update with extended follow-up data in a future publication to address these important long-term considerations.

Comments 6: Please correct the spelling mistake on line 60 ‘wich’.

Please remove one ‘a’ from line 96.

Please correct the spelling of ‘recived’ to ‘received’ throughout the text.

Response 6: Thanks for noticing these mistakes. We've corrected all of them throughout the manuscript.

4. Response to Comments on the Quality of English Language

Point 1: The English could be improved to more clearly express the research.

Response 1: Thanks for that feedback. We've thoroughly reviewed and refined the manuscript's English to enhance clarity and precision throughout the research presentation.

Reviewer 2 Report

Comments and Suggestions for Authors

Dear authors,

I have only a few brief suggestions for your report:

Line 237: Could you provide more information to explain the differences in chemotherapy cycles? How many of the initial cycles were administered? You could also mention that the combined analysis of OPRA and CA-12 reported comparable results despite the different numbers of consolidation chemotherapy cycles (DOI: 10.1016/j.ejca.2024.114291). (DOI: 10.1016/j.ejca.2024.114291).

While you provided detailed information about the treatment protocols that patients received, I am missing information about adherence. Could you report how many patients received the treatment as initially planned and how many patients required treatment adaptation? (See, for example, DOI: 10.1001/jamaoncol.2020.2394.)

On line 465, you state that the DFS is 14.7 months. Could you please clarify what you mean by this? Confidence interval? As the follow-up period is quite short, I would not recommend adding this information here.

Did you observe an association between the length of time between the end of radiotherapy and the pCR rate?

Author Response

Comments 1: Line 237: Could you provide more information to explain the differences in chemotherapy cycles? How many of the initial cycles were administered? You could also mention that the combined analysis of OPRA and CA-12 reported comparable results despite the different numbers of consolidation chemotherapy cycles (DOI: 10.1016/j.ejca.2024.114291). (DOI: 10.1016/j.ejca.2024.114291).

Response 1: Thank you for your valuable suggestions. We've updated the manuscript to include our decision-making algorithm and Figure 1. These revisions can be found in the Materials and Method section.

Comments 2: While you provided detailed information about the treatment protocols that patients received, I am missing information about adherence. Could you report how many patients received the treatment as initially planned and how many patients required treatment adaptation? (See, for example, DOI: 10.1001/jamaoncol.2020.2394.)

Response 2: Thank you for your valuable suggestions. We've updated the manuscript to include details regarding our clinical management of adverse events. These revisions can be found in the Materials and Method section.

Additionally, the paragraph "Treatment-Related Toxicities" specifies how many patients required treatment adaptation.

Comments 3: On line 465, you state that the DFS is 14.7 months. Could you please clarify what you mean by this? Confidence interval? As the follow-up period is quite short, I would not recommend adding this information here.

Response 3: We agree with your comment and have removed DFS from our conclusion.

Comments 4: Did you observe an association between the length of time between the end of radiotherapy and the pCR rate?

Response 4: Thanks for this excellent remark. In our study, surgical intervention was performed in 60 patients, with a median interval of 8 weeks (median 60.5 days, range 15-282 days) following the completion of TNT. Specifically, the 18 patients who achieved a pCR underwent surgery after a median of 106 days (range 57-164 days) from the end of radiotherapy. However, we believe that the relatively small number of patients achieving pCR, coupled with the potential confounding influence of consolidation chemotherapy, precludes a statistically robust analysis of any association between the length of time from radiotherapy completion to surgery and the pCR rate.

4. Response to Comments on the Quality of English Language

Point 1: The English is fine and does not require any improvement.

Response 1: Thanks for that feedback. We've thoroughly reviewed and refined the manuscript's English to enhance clarity and precision throughout the research presentation.

Reviewer 3 Report

Comments and Suggestions for Authors

Introduction: The introduction provides an appropriate overview of the expanding role of total neoadjuvant therapy in the management of LARC, referencing key international trials (RAPIDO, PRODIGE 23...). However, it lacks clarity regarding the clinical criteria that guide the choice among different therapeutic sequences in real-world settings, such as age, comorbidities, performance status, or surgical risk. Moreover, the biological rationale for TNT—such as early micrometastatic control, reduction of circulating tumor cells, and enhanced nodal response—is not sufficiently addressed, despite its growing relevance in recent literature as a conceptual foundation for treatment intensification. The watch-and-wait strategy, which represents a significant advancement in the multidisciplinary management of patients achieving clinical complete response, is discussed only later in the manuscript. A brief mention in the introduction would help contextualize the emerging therapeutic objectives. Including updated pCR rates from major clinical trials could further reinforce the clinical relevance of the study. Finally, the authors should highlight from the outset the need for treatment individualization in LARC, as reflected by the diversity of TNT regimens adopted across institutions and endorsed by current NCCN and ESMO guidelines.

Materials and Method: the methodology section is well-structured and adheres to standards for retrospective studies. Inclusion and exclusion criteria are clearly reported and align with those of major randomized trials. The description of chemotherapy protocols, radiotherapy details, and criteria for clinical and pathological response evaluation is thorough and consistent with international recommendations (CTCAE v5.0, RECIST v1.1, TRG). The use of advanced imaging (including PET-CT in selected cases) and the multidisciplinary tumor board assessment are commendable. However, it remains unclear whether the assignment to iCHT, cCHT, or sandwichCHT was based on a predefined institutional protocol or left to clinical discretion. The absence of a standardized decision-making algorithm limits the interpretability of intergroup comparisons and introduces potential selection bias. Furthermore, the manuscript does not report predefined criteria for managing severe adverse events or for treatment modification/interruption in cases of cumulative toxicity. A flowchart summarizing the treatment pathways would greatly enhance clarity, especially given the complexity of therapeutic sequences. The absence of QoL data is also notable, particularly given the growing emphasis on functional outcomes in modern LARC management.

Results: the results section is comprehensive and provides a detailed account of patient demographics, treatment regimens, toxicity, clinical and pathological responses, and oncologic outcomes. The distribution of patients across the three TNT strategies is clearly presented, including regimen specifics, number of cycles administered, and dose modifications. The overall pCR rate of 30% and ycCR rate of 33% are consistent with published data from pivotal trials. The absence of local recurrences at the median follow-up is a noteworthy finding. However, the median follow-up of 17 months is relatively short and may underestimate long-term DFS, which in this cohort was reported as 14.7 months. The small number of patients in the cCHT group (n=7) substantially limits statistical comparisons across treatment strategies. No formal statistical analyses are provided—no significance tests, confidence intervals, or multivariable models to assess predictors of response or progression. Additionally, data on long-term follow-up adherence, late complications, and functional outcomes (e.g., continence, QoL, sphincter preservation) are missing, despite their clinical relevance.

Discussion: the discussion is well-developed and appropriately integrates the study findings with existing literature. The authors offer a meaningful comparison with key randomized trials, particularly in terms of pCR rates, highlighting the favorable outcomes observed with the sandwichCHT approach. Treatment tolerability and real-world feasibility are appropriately emphasized. The discussion of nonoperative management (watch-and-wait) in patients with clinical complete response is balanced, although the follow-up protocol applied in these cases is not described in sufficient detail. The absence of local recurrence is acknowledged but could be better contextualized with respect to the radiotherapy techniques employed (e.g., VMAT, selective boost), which may have contributed to improved local control. Study limitations are addressed transparently, including the retrospective nature, limited follow-up, and lack of standardized treatment allocation. However, the manuscript would benefit from a broader discussion on the future role of individualized treatment strategies based on molecular markers, radiomics, or advanced imaging. Similarly, no exploratory analysis of high-risk subgroups (e.g., MRF+, N2, elderly patients) is provided, which would be valuable for the scientific community.

Author Response

Comments 1: The introduction provides an appropriate overview of the expanding role of total neoadjuvant therapy in the management of LARC, referencing key international trials (RAPIDO, PRODIGE 23...). However, it lacks clarity regarding the clinical criteria that guide the choice among different therapeutic sequences in real-world settings, such as age, comorbidities, performance status, or surgical risk. Moreover, the biological rationale for TNT—such as early micrometastatic control, reduction of circulating tumor cells, and enhanced nodal response—is not sufficiently addressed, despite its growing relevance in recent literature as a conceptual foundation for treatment intensification. The watch-and-wait strategy, which represents a significant advancement in the multidisciplinary management of patients achieving clinical complete response, is discussed only later in the manuscript. A brief mention in the introduction would help contextualize the emerging therapeutic objectives. Including updated pCR rates from major clinical trials could further reinforce the clinical relevance of the study. Finally, the authors should highlight from the outset the need for treatment individualization in LARC, as reflected by the diversity of TNT regimens adopted across institutions and endorsed by current NCCN and ESMO guidelines.

Response 1: Thank you for highlighting these important points. We've enhanced the Introduction to address your valuable feedback. Specifically, we now include:

·       A more explicit discussion of national and international guidelines recommending TNT, along with insights into the ongoing debate regarding optimal sequencing within TNT regimens.

·       Relevant pCR rates observed following TNT, strengthening the evidence base.

·       A clearer articulation of the need for individualized treatment strategies, reflecting the complex interplay of patient and treatment factors.

·       A more detailed explanation of the non-operative management (NOM) strategy, its rationale, and associated challenges.

These revisions can be found in the Introduction section.

Comments 2: the methodology section is well-structured and adheres to standards for retrospective studies. Inclusion and exclusion criteria are clearly reported and align with those of major randomized trials. The description of chemotherapy protocols, radiotherapy details, and criteria for clinical and pathological response evaluation is thorough and consistent with international recommendations (CTCAE v5.0, RECIST v1.1, TRG). The use of advanced imaging (including PET-CT in selected cases) and the multidisciplinary tumor board assessment are commendable. However, it remains unclear whether the assignment to iCHT, cCHT, or sandwichCHT was based on a predefined institutional protocol or left to clinical discretion. The absence of a standardized decision-making algorithm limits the interpretability of intergroup comparisons and introduces potential selection bias. Furthermore, the manuscript does not report predefined criteria for managing severe adverse events or for treatment modification/interruption in cases of cumulative toxicity. A flowchart summarizing the treatment pathways would greatly enhance clarity, especially given the complexity of therapeutic sequences. The absence of QoL data is also notable, particularly given the growing emphasis on functional outcomes in modern LARC management.

Response 2: Thank you for your valuable suggestions. We've updated the manuscript to include our decision-making algorithm (Figure 1) and details regarding our clinical management of adverse events. These revisions can be found in the Materials and Method section.

Comments 3: the results section is comprehensive and provides a detailed account of patient demographics, treatment regimens, toxicity, clinical and pathological responses, and oncologic outcomes. The distribution of patients across the three TNT strategies is clearly presented, including regimen specifics, number of cycles administered, and dose modifications. The overall pCR rate of 30% and ycCR rate of 33% are consistent with published data from pivotal trials. The absence of local recurrences at the median follow-up is a noteworthy finding. However, the median follow-up of 17 months is relatively short and may underestimate long-term DFS, which in this cohort was reported as 14.7 months. The small number of patients in the cCHT group (n=7) substantially limits statistical comparisons across treatment strategies. No formal statistical analyses are provided—no significance tests, confidence intervals, or multivariable models to assess predictors of response or progression. Additionally, data on long-term follow-up adherence, late complications, and functional outcomes (e.g., continence, QoL, sphincter preservation) are missing, despite their clinical relevance.

Response 3: We agree with this comment. The present follow-up duration precludes more comprehensive or definitive long-term analyses and results. We plan to provide an update with extended follow-up data in a future publication.

Comments 4: the discussion is well-developed and appropriately integrates the study findings with existing literature. The authors offer a meaningful comparison with key randomized trials, particularly in terms of pCR rates, highlighting the favorable outcomes observed with the sandwichCHT approach. Treatment tolerability and real-world feasibility are appropriately emphasized. The discussion of nonoperative management (watch-and-wait) in patients with clinical complete response is balanced, although the follow-up protocol applied in these cases is not described in sufficient detail. The absence of local recurrence is acknowledged but could be better contextualized with respect to the radiotherapy techniques employed (e.g., VMAT, selective boost), which may have contributed to improved local control. Study limitations are addressed transparently, including the retrospective nature, limited follow-up, and lack of standardized treatment allocation. However, the manuscript would benefit from a broader discussion on the future role of individualized treatment strategies based on molecular markers, radiomics, or advanced imaging. Similarly, no exploratory analysis of high-risk subgroups (e.g., MRF+, N2, elderly patients) is provided, which would be valuable for the scientific community.

Response 4: Thanks for your valuable input! We've made the following updates to the manuscript:

·       We've added details regarding our NOM program follow-up protocol in the Methods and Materials section;

·       To emphasize the critical role of radiotherapy, we've highlighted the impact of advanced radiotherapy techniques and doses on local control. These revisions can be found in the Discussion section;

·       We also incorporated a sentence underscoring the importance of identifying predictive biomarkers for personalizing therapeutic approaches. This addition is located in the Discussion section.

4. Response to Comments on the Quality of English Language

Point 1: The English could be improved to more clearly express the research.

Response 1: Thanks for that feedback. We've thoroughly reviewed and refined the manuscript's English to enhance clarity and precision throughout the research presentation.

Reviewer 4 Report

Comments and Suggestions for Authors

The 70 pts. treated with a large variety of protocols obviously are well analyzed. The sections of the paper by far are well written, including the conclusions.

I have several questions/remarks related to the lines in the paper:

l 5-12:  please add the numbers to the names of the authors

l 23, l26: Please explain exactly the term "TNT". There are several authors, including the initial female author, who understand "TNT" as the prcedure to avoid surgery (=conservative treatment of RC). 

l 32: please explain "CRT" in the abstract

l 36: Obviously in your hospital you use neo CRT+ surg. + adj CT as the preferred  concept with good results!

l 53: "NOM" usually is the primary aim of TNT? - So, please explain the differences between TNT and NOM, for example after line 94.? In your strategy, concervative treatment is not a planned option.

l 120: What is "bladder preparion"?

Table 1: Male+Female = 60 pts.! You are writing your report on 70 pts.. please control all your numbers!

l  234: Determining DPYD is very interesting (we have done this in a phase III decision aiding trial long before...). Unfortunately, you do not report any results/correlations to responses/survival.

l 323: Again, yout intention was combined treatment with CRT + Surg.. Your R0 resection rate of 100% is extremely good and really pathologically confirmed in all 60 resected specimen?

As to  your clinical results with DFS(OAS, reported in lines 364-379, it would be nice to demonstrate in a figure. DFS of 14.7 months is rather low and difficult to understand with only 17% experiencing PD.

l 381: Again, explain exactly "TNT" and prevent, that TNT is misunderstood (???) as a procedure with the primary aim to avoid surgery.

l 465: DFS of 14.7 months: Is this really encouring?

Overall, the paper is interesting and gives additional evidence, that your preferred procedure, the sandwich treatment including LCRT, is very acceptable. Has this procedure meanwhile been included in any guide line in Europe?  

Please decide wht and how to correct. Than the paper may be published.

Comments on the Quality of English Language

The language to my opinion is good. It never hurts, if language is examined.

Author Response

Comments 1: The 70 pts. treated with a large variety of protocols obviously are well analyzed. The sections of the paper by far are well written, including the conclusions. I have several questions/remarks related to the lines in the paper:

5-12:  please add the numbers to the names of the authors

Response 1: Thanks for this comment. We hope the automated matching process between our author details and the manuscript draft will resolve the numbering issue in the final version.

Comments 2: 23, 26: Please explain exactly the term "TNT". There are several authors, including the initial female author, who understand "TNT" as the procedure to avoid surgery (=conservative treatment of RC).

Response 2: Thanks for this comment. We've clarified in the Abstract that TNT refers to the administration of both chemoradiotherapy (CRT) and systemic chemotherapy (CHT) prior to surgical resection. These revisions can be found in the Abstract section.

Comments 3: 32: please explain "CRT" in the abstract.

Response 3: Thanks for catching that oversight. We've now explained CRT in the Abstract.

Comments 4: 36: Obviously in your hospital you use neo CRT+ surg. + adj CT as the preferred concept with good results!

Response 4: Yes, neoadjuvant chemoradiotherapy followed by surgery and adjuvant chemotherapy was indeed our preferred strategy before the adoption of TNT, and it yielded favorable results in our institution.

Comments 5: 53: "NOM" usually is the primary aim of TNT? - So, please explain the differences between TNT and NOM, for example after line 94.? In your strategy, concervative treatment is not a planned option.

Response 5: Thank you for highlighting these important points. We've enhanced the Introduction to address your valuable feedback. Specifically, we now include a more detailed explanation of the non-operative management (NOM) strategy, its rationale, and associated challenges. These revisions can be found in the Introduction section.

Comments 6: 120: What is "bladder preparation"?

Response 6: "Bladder preparation" in the context of pelvic radiotherapy refers to ensuring a comfortably full bladder. This is crucial for treatment accuracy and minimizing side effects. Our protocol advises patients to drink 1.5-2 liters of water daily. Before each radiotherapy session, patients are instructed to empty their bladder and then consume 500 milliliters of water, with treatment commencing 30 minutes thereafter.

Comments 7: Table 1: Male+Female = 60 pts.! You are writing your report on 70 pts.. please control all your numbers!

Response 7: Thanks for pointing this out. We've reviewed the numbers, and Table 1 correctly reports 49 males and 21 females, totaling 70 patients.

Comments 8: 234: Determining DPYD is very interesting (we have done this in a phase III decision aiding trial long before...). Unfortunately, you do not report any results/correlations to responses/survival.

Response 8: Thanks for this insightful comment regarding DPYD genotyping. While DPYD determination is indeed a valuable aspect of patient management, our current median follow-up of 17 months is too short to provide statistically robust analyses of its correlations with treatment responses or survival outcomes. We aim to address these important considerations in a future publication with extended follow-up data.

Comments 9: Again, your intention was combined treatment with CRT + Surg. Your R0 resection rate of 100% is extremely good and really pathologically confirmed in all 60 resected specimen?

Response 9: Yes, our R0 resection rate of 100% was indeed pathologically confirmed in all 60 resected specimens. We are extremely pleased with these results and commend our surgical team.

Comments 10: As to your clinical results with DFS (OAS, reported in lines 364-379, it would be nice to demonstrate in a figure. DFS of 14.7 months is rather low and difficult to understand with only 17% experiencing PD.

465: DFS of 14.7 months: Is this really encouring?

Response 10: We agree with your comment and have removed DFS from our conclusion.

Comments 11: Again, explain exactly "TNT" and prevent, that TNT is misunderstood (???) as a procedure with the primary aim to avoid surgery.

Response 11: You're absolutely right to highlight this potential for misunderstanding. We agree that while TNT aims for optimal oncologic outcomes, non-operative management NOM isn't always the primary intent from the outset. Often, NOM emerges as a viable strategy only after the completion of both chemotherapy and radiotherapy (administered with a primary "neoadjuvant" intent) and subsequent restaging reveals a clinical complete response. The current challenge, therefore, is to accurately identify patients who will achieve a pathological complete response after neoadjuvant treatment, enabling us to safely offer NOM without compromising oncologic safety. In the Introduction section we’ve emphasized that NOM should be an integral part of the treatment discussion for LARC.

Comments 12: Overall, the paper is interesting and gives additional evidence, that your preferred procedure, the sandwich treatment including LCRT, is very acceptable. Has this procedure meanwhile been included in any guide line in Europe?  Please decide what and how to correct. Than the paper may be published.

Response 12: Thanks for the positive report. No, our "sandwich" treatment approach is not currently included in any European guidelines. To our knowledge, there are also no other institutions that have yet investigated this specific sequencing strategy. We hope our paper will stimulate further research into this promising regimen.

4. Response to Comments on the Quality of English Language

Point 1: The English could be improved to more clearly express the research.

Response 1: Thanks for that feedback. We've thoroughly reviewed and refined the manuscript's English to enhance clarity and precision throughout the research presentation.

Round 2

Reviewer 1 Report

Comments and Suggestions for Authors

Approved. Satisfied with the answers of major questions

Reviewer 3 Report

Comments and Suggestions for Authors

The revisions are enough.